# The impact of properly diagnosed sarcopenia on postoperative outcomes after gastrointestinal surgery: A systematic review and meta-analysis

Leonardo Zumerkorn Pipek[1], Carlos Guilherme Baptista[2], Rafaela Farias Vidigal Nascimento[3], João Victor Taba[1], Milena Oliveira Suzuki[1], Fernanda Sayuri do Nascimento[1], Diego Ramos Martines[1], Fernanda Nii[1], Leandro Ryuchi Iuamoto[4], Luiz Augusto Carneiro-D'Albuquerque[2], Alberto Meyer[2]*, Wellington Andraus[2]

1 Faculdade de Medicina FMUSP, Universidade de Sao Paulo, Sao Paulo, São Paulo, Brazil, 2 Departamento de Gastroenterologia, Hospital das Clínicas, HCFMUSP, São Paulo, Brazil, 3 Faculdade de Medicina do ABC - Centro Universitário Saúde ABC, Santo André, São Paulo, Brasil, 4 Department of Orthopaedics and Traumatology, Center of Acupuncture, University of Sao Paulo School of Medicine, Sao Paulo, Brazil

* alberto.meyer@usp.br

**Data Availability Statement:** All relevant data are within the paper and its Supporting Information files.

## Abstract

### Background

Sarcopenia is defined as the loss of muscle mass combined with loss of muscle strength, with or without loss of muscle performance. The use of this parameter as a risk factor for complications after surgery is not currently used. This meta-analysis aims to assess the impact of sarcopenia defined by radiologically and clinically criteria and its relationship with complications after gastrointestinal surgeries.

### Materials and methods

A review of the literature was conducted in accordance with the Preferred Reporting Items for Systematic Reviews and Meta-Analyses (PRISMA) guidelines (PROSPERO registration number: CRD42019132221). Articles were selected from the PUBMED and EMBASE databases that adequately assessed sarcopenia and its impact on postoperative complications in gastrointestinal surgery patients. Pooled estimates of pre-operative outcome data were calculated using the odds ratio (OR) and 95% confidence interval (CI). Subgroup analysis were performed to assess each type of surgery.

### Results

The search strategy returned 1323, with 11 studies meeting the inclusion criteria. A total of 4265 patients were analysed. The prevalence of sarcopenia between studies ranged from 6.8% to 35.9%. The meta-analysis showed an OR for complications after surgery of 3.01 (95% CI 2.55–3.55) and an OR of 2.2 (95% CI 1.44–3.36) for hospital readmission (30 days).

**Funding:** The author(s) received no specific funding for this work.

**Competing interests:** The authors have declared that no competing interests exist.

## Conclusion

Sarcopenia, when properly diagnosed, is associated with an increase in late postoperative complications, as well as an increase in the number of postoperative hospital readmissions for various types of gastrointestinal surgery. We believe that any preoperative evaluation should include, in a patient at risk, tests for the diagnosis of sarcopenia and appropriate procedures to reduce its impact on the patient's health.

## 1. Introduction

The steady growth in the number of surgical procedures is a consequence of increased life expectancy and better surgical techniques [1]. Gastrointestinal surgeries are among the most performed surgical procedures, especially those related to the resection of tumors [2]. It is up to the medical team, together with the patient, to analyse the risks of each surgery and decide which is the best therapeutic option [1]. In this context, tools have been developed to assist in this decision, such as the Charlson Comorbidity Index, so the factors that contribute to the success of the procedure can be evaluated [3].

These factors can be divided into those intrinsic to the disease process, such as the pathophysiological characteristics of a tumor and its staging, and those related to the patient. In relation to the latter, some of the factors that are very prevalent in the population, such as sarcopenia, have not yet been included in the analysis tools due to the scarcity of studies that accurately determine its impact on surgical procedures.

Sarcopenia is defined as the loss of muscle mass combined with loss of muscle strength, with or without loss of muscle performance [4,5]. Evidence indicates that it starts in the fifth decade of life and has a linear progression, being characterized as primary [6,7]. There is also the secondary form that results from a pathological process, as in the case of chronic-degenerative diseases, malnutrition, and chronic inflammation [8]. In addition, some studies emphasize the importance of vitamin D deficiency [9] and insulin resistance [10] in the pathogenesis of the disease. Considering that muscle mass represents approximately 60% of the body mass and is metabolically active, it is expected that sarcopenia affects homeostasis substantially [11]. From a histological point of view, there is a reduction in the size and number of muscle cells, especially type 2, and infiltration of fibrosis and fat [12].

All of these changes have a great impact on the lives of individuals, from decreased ability to perform daily activities [13], falls, increased prevalence of frailty syndrome [14–16] and, especially, in the state of acute and chronic diseases, increased mortality and comorbidity [17].

Given the importance of sarcopenia as one of the determining factors to the success of surgery, it is essential that there are precise ways of diagnosing it. Many articles use only muscle mass to diagnose sarcopenia, mainly due to the fact it is easy to obtain such data from imaging tests that would already be performed on patients regardless. Therefore, one of the most common methods used is the cross-sectional area of skeletal muscles on abdominal CT at the level of the third lumbar vertebra (L3) [18–22]. However, according to The European Working Group on Sarcopenia in Older People (EWGSOP), this data alone is not enough: in addition to the loss of muscle mass, it is necessary to have an assessment of strength and / or physical performance for it to be verified sarcopenia[4]. Regarding the measurement of muscle strength, the most used and validated form is the hand grip exam—non-dominant hand grip strength [23,24].

This review aims to analyse articles that have assessed the presence of sarcopenia in adult patients, according to the recommended criteria, and to relate this factor to the outcome of gastrointestinal surgeries.

## 2. Materials and methods

This systematic review and meta-analysis was carried out in accordance with the Preferred Reporting Items for Systematic Reviews and Meta-Analyses (PRISMA) [25].

The elaboration of the scientific question was based on the PICO strategy [26] considering:

P—Patient/Problem–adult patients undergoing gastrointestinal surgery;

I–Intervention/ Prognosis Factor—presence or absence of a diagnosis of sarcopenia, according to internationally recognized methods;

C–Control/ Comparison—there is no standard intervention to be considered in this study;

O–Outcome—influence of sarcopenia on early and late postoperative results.

### 2.1 Eligibility criteria

The following criteria were adopted for the inclusion of studies in this review:

1. They must report results of surgical interventions of a gastrointestinal nature (e.g. hernia, colectomy, GI oncology) regarding the presence of the diagnosis of sarcopenia, in patients aged 18 years or older.

2. Sarcopenia must be related to age (primary) or underlying disease (secondary).

3. The description of the post-surgical results must have at least one of the following data: postoperative complications, infectious complications, complications of a clinical nature, and mortality data. They can be early, late or both.

### 2.2 Exclusion criteria

The exclusion criteria were:

1. Diagnosis of sarcopenia based on purely laboratory or imaging criteria (for example, tomography only).

2. Sarcopenia secondary to intervention (e.g. post-transplant, post-chemotherapy).

3. Studies involving surgeries of a different nature than gastrointestinal

4. Letters to editors, incomplete unpublished articles and research protocols.

### 2.3 Types of studies

Retrospective and prospective cohort studies that compared gastrointestinal surgery outcomes in sarcopenic and non-sarcopenic patients were included.

### 2.4 Data search strategy and literature review

The study was registered in the PROSPERO database under number CRD42019132221. Two independent reviewers performed a systematic search of the PubMed and EMBASE databases; only articles in the English language were searched. The period sought database inception to the 3rd of April 2020. The systematic review followed the PRISMA guidelines (Preferred Reporting Items for Systematic Reviews and Meta-Analysis) [25].

Using the PubMed search tool, we selected the MeSH terms of the most relevant publications to conduct a new search in order to obtain more articles that could potentially be included in this review.

The keywords used for were: sarcopenia, muscle wasting, gastrointestinal surgery, complications, outcomes, and mortality.

PUBMED: ("Sarcopenia"[Mesh] OR "Muscular Atrophy"[Mesh] OR "Frailty"[Mesh]) AND ("Digestive System Surgical Procedures"[Mesh] OR "gastrointestinal surgery") AND ("Patient Outcome Assessment"[Mesh] OR "Postoperative Complications"[Mesh] OR "Mortality"[-Mesh]): → 167.

EMBASE: ('sarcopenia'/exp OR 'sarcopenia' OR 'muscular atrophy'/exp OR 'muscular atrophy' OR 'frailty'/exp OR 'frailty') AND ('digestive system surgical procedures'/exp OR 'digestive system surgical procedures' OR 'gastrointestinal surgery'/exp OR 'gastrointestinal surgery') AND ('patient outcome assessment'/exp OR 'patient outcome assessment' OR 'postoperative complications'/exp OR 'postoperative complications' OR 'mortality'/exp OR 'mortality') → 1245.

## 2.5 Selection of studies and data extraction

Based on the criteria mentioned above, the headlines and abstracts of the studies were initially screened by two independent reviewers. Relevant articles were assessed for methodology, especially for the method of diagnosing sarcopenia. After this second screening, the remaining articles were read in full to define their inclusion in the review. Disagreeing cases were resolved with a third reviewer (AM).

The main data for each article included in the review were extracted by a reviewer (LP). The following were tabled: article headline, authors, year of publication, country of origin, basic characteristics of the study populations, study design, study objectives, sample size, inclusion and exclusion criteria, gender relationship, prevalence of sarcopenia, method used to diagnose sarcopenia, type of surgery, primary and secondary outcomes regarding the presence or absence of sarcopenia. The data were checked by another reviewer (CGB).

## 2.6 Data quality analysis

The studies included in the review were assessed for the quality of the data presented using the MINORS tool (Methodological Index for Non-Randomized Studies) [27]. The instrument consists of eight items for non-comparative studies, plus four other items for comparative studies. The LP and CGB reviewers performed the methodological quality analysis of the studies as described, and inconclusive cases were assessed by a third reviewer (AM).

## 2.7 Statistical analysis

Heterogeneity measures were used to evaluate the studies, Higgins I2 measure, which indicates the variation in the association measure attributable to heterogeneity. The Cochran test under the null hypothesis that there is no heterogeneity was calculated. Fixed effects regression model was used for the data analysis. The overall effect was assessed under the null hypothesis that the odds ratio (OR) = 1 (there is no effect between the exposure levels).

The graphs of forest plot and funnel plot (to assess publication bias) were performed.

The subgroup analysis was performed based on the type of surgery in each study.

The analyses were performed using the statistical software RStudio Team (2015). RStudio: Integrated Development for R. RStudio, Inc., Boston, MA

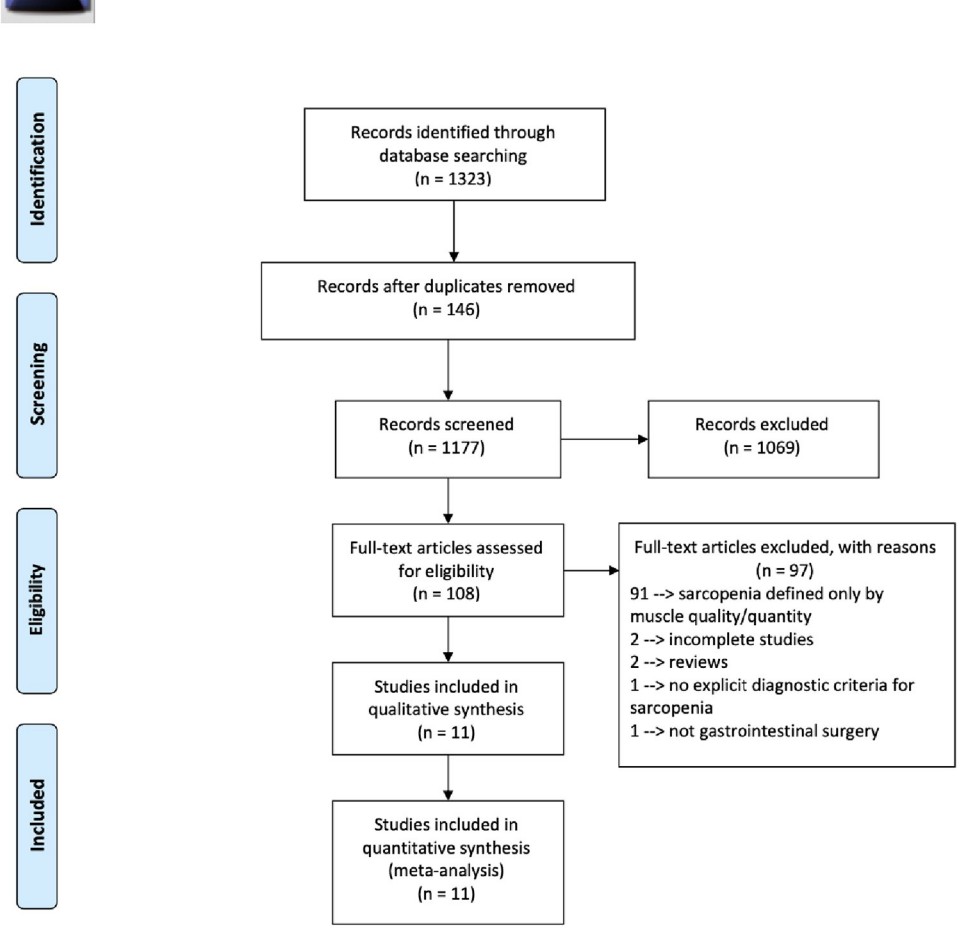

**Fig 1. Data search flow.**

## 3. Results

### 3.1 Selected studies

The search strategy resulted in 1323 articles. After removing duplicate articles, 1177 remained. These underwent initial screening, focusing on headlines and abstracts. After this initial screening, 108 articles remained for evaluation of the full text. Of these, 11 articles had the necessary characteristics for inclusion [28–38]. (Fig 1).

### 3.2 Population characteristics

Of the selected studies, eight involved patients with gastric cancer [29–34,36,37], two with colorectal cancer [28,38], and one with esophageal cancer [35]. The total number of subjects evaluated was 4265, between sarcopenic and non-sarcopenic. All studies associated imaging with functional testing for the diagnosis of sarcopenia, and all originated in two countries: Japan [31,35,37] and China [28–30,32–34,36,38]. Table 1 summarizes the population characteristics of each study.

**Table 1. Population characteristics of the 11 selected studies.**

| Author, Year | Study design | Type of surgery | N total | Sarcopenia prevalence | Age | | | | Sex ratio: M/F | | | Data collection period | Country |
|---|---|---|---|---|---|---|---|---|---|---|---|---|---|
| | | | | | Sarcopenic | | Non Sarcopenic | | Sarcopenic | Non Sarcopenic | | | |
| Huang 2015 | Prospective | Colorectal Cancer | 142 | 12.00% | 74.59 (6.65) | • | 60.32 (12.24) | • | 1.83 | 1.60 | | August 2014 and February 2015 | China |
| Fukuda 2016 | Retrospective | Gastric Cancer | 99 | 21.20% | 78 (67–85) | † | 75 (66–91) | † | 9.50 | 1.52 | | July 2012 and January 2015 | Japan |
| Wang 2016 | Prospective | Gastric Cancer | 255 | 12.50% | 74.66 (6.80) | • | 63.77 (10.66) | • | 4.33 | 2.78 | | August 2014 to March 2015 | China |
| Huang 2017 | Prospective | Gastric Cancer | 470 | 16.80% | 63 (4) | † | 74 (10) | † | 2.95 | 3.32 | | August 2014 to December 2015 | China |
| Lou 2017 | Prospective | Gastric Cancer | 206 | 6.80% | 74.78 (5.08) | • | 63.27 (9.93) | • | 1.80 | 3.80 | | August 2014 to December 2015 | China |
| Zhou 2017 | Prospective | Gastric Cancer | 240 | 28.80% | 76 (6.5) | • | 71 (7) | • | 3.06 | 4.18 | | August 2014 to December 2015 | China |
| Kawamura 2018 | Retrospective | Gastric Cancer | 951 | 11.70% | 76 (65–90) | † | 72.5 (65–87) | † | 1.64 | 2.37 | | July 2003 and June 2011 | Japan |
| Makiura 2018 | Prospective | Esophageal Cancer | 113 | 31.60% | 69 (65–75) | † | 64 (61–70) | † | 3.86 | 7.37 | | June 2011 to November 2014 | Japan |
| Chen 2018 | Prospective | Colorectal Cancer | 376 | 24.50% | 70.73 (12.61) | • | 59.20 (13.0) | • | 2.00 | 2.04 | | July 2014 to February 2017 | China |
| Ma 2019 | Prospective | Gastric Cancer | 545 | 7.30% | 70.63 (11.13) | • | 61.93 (10.20) | • | 1.67 | 3.50 | | August 2014 to December 2017 | China |
| Zhuang 2019 | Prospective | Gastric Cancer | 883 | 35.90% | 73.54 (7.50) | • | - | | 2.40 | - | | August 2014 to February 2018 | China |

† mean (sd).

· median (IQR).

## 3.3 Quality assessment using the MINORS score

Table 2 describes the score received by each study in the quality assessment[28–38]. Through the MINORS scale, up to 24 points can be awarded for comparative studies; the score of the studies included in this review ranged from 16 to 22. The main factors that led to the loss of

**Table 2. ORS score for quality assessment (A). Criteria used for the assessment (B).**

| A | | | | | | | | | | | | | | B |
|---|---|---|---|---|---|---|---|---|---|---|---|---|---|---|
| Study | 1 | 2 | 3 | 4 | 5 | 6 | 7 | 8 | 9 | 10 | 11 | 12 | **TOTAL** | Criteria: |
| Huang 2015 | 2 | 2 | 2 | 2 | 2 | 1 | 2 | 0 | 2 | 2 | 1 | 2 | **20** | 1—Clearly stated aim |
| Fukuda 2016 | 2 | 2 | 0 | 2 | 0 | 1 | 2 | 0 | 2 | 2 | 2 | 2 | **17** | 2—Inclusion of consecutive patients |
| Wang 2016 | 2 | 2 | 2 | 2 | 2 | 2 | 2 | 0 | 2 | 2 | 2 | 2 | **22** | 3—Prospective collection of data |
| Huang 2017 | 1 | 2 | 2 | 2 | 2 | 1 | 2 | 0 | 2 | 2 | 2 | 2 | **20** | 4—Endpoints appropriate to the aim of the study |
| Lou 2017 | 2 | 2 | 2 | 2 | 2 | 1 | 2 | 0 | 2 | 2 | 1 | 2 | **20** | 5—Unbiased assessment of the study endpoint |
| Zhou 2017 | 2 | 2 | 2 | 2 | 2 | 2 | 2 | 0 | 2 | 2 | 2 | 2 | **22** | 6—Follow-up period appropriate to the aim of the study |
| Kawamura 2018 | 1 | 2 | 0 | 2 | 0 | 2 | 2 | 0 | 2 | 2 | 2 | 2 | **17** | 7—Loss to follow up less than 5% |
| Makiura 2018 | 2 | 2 | 2 | 2 | 0 | 2 | 0 | 0 | 2 | 2 | 2 | 2 | **18** | 8—Prospective calculation of the study size |
| Chen 2018 | 2 | 2 | 2 | 2 | 2 | 2 | 2 | 0 | 2 | 2 | 2 | 2 | **22** | 9—An adequate control group |
| Ma 2019 | 2 | 2 | 2 | 2 | 2 | 2 | 2 | 0 | 2 | 2 | 1 | 2 | **21** | 10—Contemporary groups |
| Zhuang 2019 | 2 | 2 | 2 | 2 | 0 | 2 | 2 | 0 | 2 | 2 | 2 | 2 | **20** | 11—Baseline equivalence of groups |
| | | | | | | | | | | | | | | 12—Adequate statistical analyses |

| Classification | Criteria |
|---|---|
| EWGSOP2 | Probable sarcopenia is identified by Criterion 1. Diagnosis is confirmed by additional documentation of Criterion 2. If Criteria 1, 2 and 3 are all met, sarcopenia is considered severe. |
| | (1) Low muscle strength |
| | (2) Low muscle quantity or quality |
| | (3) Low physical performance |
| AWGS | Similar to EWGSOP2 in the criteria used, but |
| | (1) measuring both muscle strength (handgrip strength) and physical performance (usual gait speed) as the screening test |
| | (2) cutoff values of these measurements in Asian populations may differ |
| IWGS | Low whole body or appendicular fat free mass in combination with poor physical functioning |
| FNIH | Weakness (grip strenght) and low muscle mass |

**Fig 2. Most used classifications for sarcopenia.** EWGSOP2 and AWGS form the criteria selected for inclusion in this systematic review and meta-analysis.

points were a very short follow-up period regarding the objectives of the study, and the retrospective design. None of the articles performed a prospective calculation of the sample size.

## 3.4 Criteria for the diagnosis of sarcopenia

The most used classifications for the diagnosis of sarcopenia that do not use only imaging criteria are AWGS [39], EWGSOP2 [40], IWGS [41] e FNIH[42] (Fig 2). However, studies show that the criteria EWGSOP2 and AWGS are the ones that best rate sarcopenia [43]. Thus, these were the criteria for the inclusion of articles. The diagnosis was made after confirmation of reduced muscle mass combined with loss of strength and / or muscle performance. The method for assessing muscle strength in all studies was the hand-grip; muscle performance assessment was performed in all studies, except one [37]. The evaluation was made through the usual walking speed, with distances ranging between 4m and 6m. The most widely used method for assessing muscle mass was tomography, with assessment of the skeletal muscle index at the level of the third lumbar vertebra and specific cutoff points for each sex [28–30,32–34,36,38]. Two studies used bioimpedance to calculate the appendicular skeletal muscle index [31,35]; one study used anthropometric measurements to calculate the muscular circumference of the arm [37].

With these criteria, the prevalence of sarcopenia between studies ranged from 6.8% to 35.9%. The data related to the sarcopenia diagnostic method are summarized in Table 3.

## 3.5 Outcomes

**3.5.1 Sarcopenia and postoperative complications.** In all 11 studies [28–38] a link was found between sarcopenia and postoperative complications. To exclude the influence of other common factors in the sarcopenic population, univariate and multivariate analyses were used. The presence of complications was studied based on the Clavien–Dindo classification. Two articles [37,38] considered a grade 2 or higher in the analyses. Three articles [31–33] separated their analysis considering a grade 2 or higher and a grade 3 or higher. Five articles [28–30,34,36] divided and specified in each of the grades 2, 3, 4 and 5. One of them [35] divided into all grades but did not specify the complications.

Another analysis of great interest refers to the division of complications into surgery and non-surgical [31–33,37,38]. Chen et al. (2018) [38] found significant differences in the two categories. Some authors [31–33,37] found differences only in non-surgical complications.

**Table 3. Methods used for the diagnosis of sarcopenia.**

| Author, Year | Muscle Mass Evaluation | Cut-off points | Strength Evaluation | Cut-off points | Performance evaluation | Cut-off points | Sarcopenia Diagnosis Criteria |
|---|---|---|---|---|---|---|---|
| Huang 2015 | Sex-specific L3 skeletal muscle index (CT images) | SMI < 36 cm2/m2 for men, < 29 cm2/m2 for women | Hand-grip | < 26kg men, < 18kg women | 6m usual gait speed | <0.8m/s | EWGSOP, AWGS |
| Fukuda 2016 | Bioimpedance analysis—skeletal muscle mass index | SMI < 8.87 kg/m2 for men, < 6.42 kg/m2 for women | Hand-grip | < 30kg men, < 20kg women | 4m usual gait speed | <0.8m/s | EWGSOP |
| Wang 2016 | Sex-specific L3 skeletal muscle index (CT images) | SMI < 36 cm2/m2 for men, < 29 cm2/m2 for women | Hand-grip | < 26kg men, < 18kg women | 6m usual gait speed | <0.8m/s | EWGSOP, AWGS |
| Huang 2017 | Sex-specific L3 skeletal muscle index (CT images) | SMI < 40.8 cm2/m2 for men, 34.9 cm2/m2 for women | Hand-grip | < 26kg men, < 18kg women | 6m usual gait speed | <0.8m/s | EWGSOP |
| Lou 2017 | Sex-specific L3 skeletal muscle index (CT images) | SMI < 40.8 cm2/m2 for men, 34.9 cm2/m2 for women | Hand-grip | < 26kg men, < 18kg women | 6m usual gait speed | <0.8m/s | AWGS |
| Zhou 2017 | Sex-specific L3 skeletal muscle index (CT images) | SMI < 40.8 cm2/m2 for men, 34.9 cm2/m2 for women | Hand-grip | < 26kg men, < 18kg women | 6m usual gait speed | <0.8m/s | EWGSOP, AWGS |
| Kawamura 2018 | Anthropometry—Arm Muscle Area | Muscle mass < 20th sex-specific percentile | Hand-grip | < 26kg men, < 18kg women | Not obtained | Not obtained | AWGS |
| Makiura 2018 | Multi- frequency bioelectrical impedance with eight electrodes | Appendicular skeletal muscle mass divided by squared height. < 7 kg/m2 for men, < 5.7kg/m2 for women | Hand-grip | < 26kg men, < 18kg women | 4m usual gait speed | <0.8m/s | AWGS |
| Chen 2018 | Sex-specific L3 skeletal muscle index (CT images) | SMI < 40.8 cm2/m2 for men, < 34.9 cm2/m2 for women | Hand-grip | < 26kg men, < 18kg women | 6m usual gait speed | <0.8m/s | AWGS |
| Ma 2019 | Sex-specific L3 skeletal muscle index (CT images) | SMI < 40.8 cm2/m2 for men, < 34.9 cm2/m2 for women | Hand-grip | < 26kg men, < 18kg women | 6m usual gait speed | <0.8m/s | EWGSOP |
| Zhuang 2019 | Sex-specific L3 skeletal muscle index (CT images) | SMI < 40.8 cm2/m2 for men, < 34.9 cm2/m2 for women | Hand-grip | < 26kg men, < 18kg women | 6m usual gait speed | <0.8m/s | EWGSOP |

In addition to the Clavien–Dindo classification, some articles used other parameters to perform the analysis: Kawamura et al. (2018) [37] investigated the occurrence of sepsis, and came to the conclusion that there is a relationship between this condition and sarcopenia.

It is worth mentioning that Huang et al. (2017) and Zhou et al. (2017) [32,33] did not find a significant difference when considering grade 3 or higher. This suggests that the relationship would occur only when considering complications in general. Fukuda et al. (2016) [31] found only significant differences only in the most severe groups (grade 3 or higher) when analyzing complications in general.

One of the studies [28] also compared the exclusive use of decreased muscle mass as a risk factor and concluded that this is not a good indicator, and proper sarcopenia definition should be used to obtain consistent results.

Some articles also investigated the number of postoperative days in the hospital as an indirect assessment of complications [28,29,31–38]. Some of them [32,33,35,36,38] found a significant difference between the sarcopenic and control groups.

**3.5.2 Sarcopenia and hospital readmission rate.** From the analysis of the 30-day hospital readmission rate, all articles found a positive relation [28,29,32–34,36,38], however only one of

them was statistically significant [34]. On the other hand, one article [35] analyzed the 90-day hospital readmission rate, and showed evidence that there is a significant difference in sarcopenics.

**3.5.3 Sarcopenia and mortality.**   Unlike postoperative complications, sarcopenia does not appear to be associated with 30-day or 3-month mortality. Articles [28,31–34,37,38] did not find differences in 30-day mortality.

However, when considering long-term mortality, sarcopenia can have an impact on patients' lives.

The survival study [30] identified sarcopenia as an independent risk factor for survival.

## 3.6 Meta-analysis

**3.6.1 Sarcopenia and complications.**   The meta-analysis corresponds to the assessment of sarcopenia as a risk factor for complications in patients undergoing digestive tract surgery.

After review, 11 publications were considered that reported the presence of sarcopenia and postoperative complications. The odds ratio (OR) values and their respective 95% confidence intervals (95% CI) were calculated.

In Fig 3, the forest plot is shown. Of the 11 publications, 10 of them found the presence of sarcopenia as a risk factor for post-surgical complications (p <0.05). One study had no statistical evidence to reject the hypothesis that the OR value was different from 1 [31]. The global OR value was 3.0 (95% CI 2.53–3.55). The measure of heterogeneity $I^2$ (Higgins heterogeneity measure) was 16%, a value considered as low heterogeneity. According to Cochran's Q heterogeneity test, the sample evidence did not allow us to reject the null hypothesis of non-heterogeneity (p = 0.29).

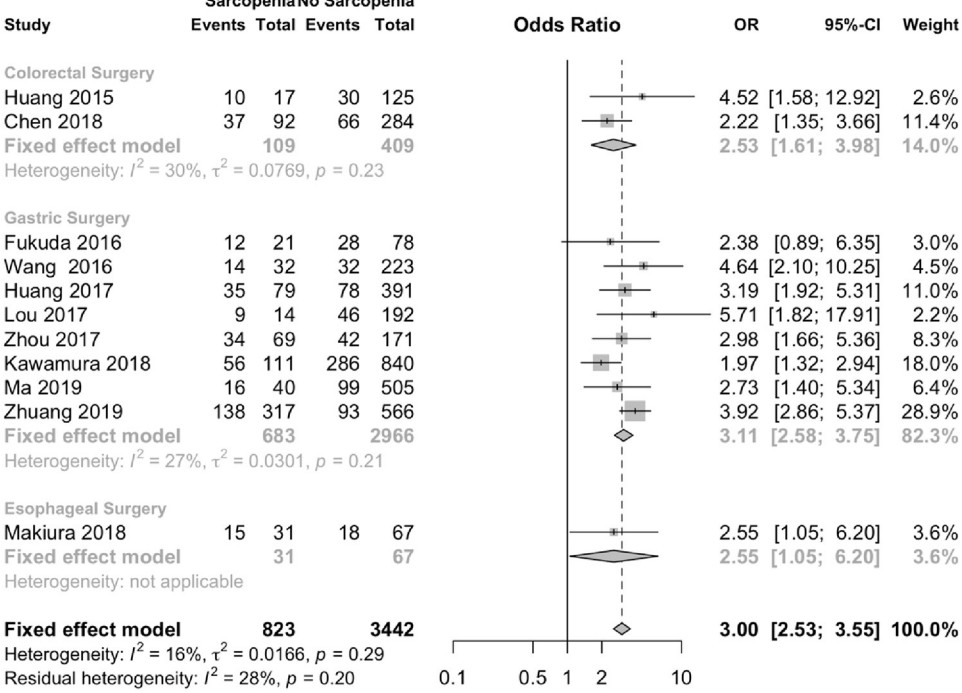

**Fig 3. Forest plot of the publications analysed in relation to sarcopenia and post-surgical complications.**

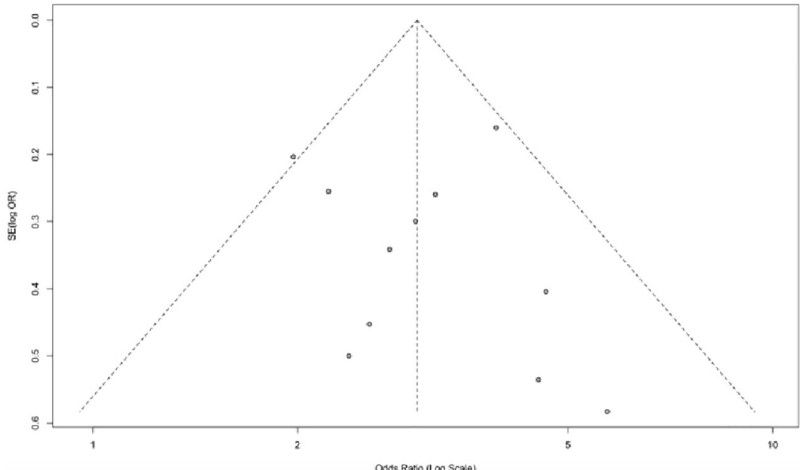

**Fig 4. Publication bias for studies that assessed sarcopenia as a factor associated with post-surgical complications.**

The publication bias was assessed using the funnel plot (Fig 4). The graph represents each of the studies with the value of association and effect measures.

The subgroup analysis for complications separated by type of surgery showed similar results. In gastrectomy studies [29–34,36,37] a total of 3649 patients were analysed. The global OR value was 3.09 (95% CI 2.44–3.92). Regarding the analysis of colorectal surgery studies [28,38], a total of 518 patients were analysed. The global OR value was 2.71 (95% CI 1.45–5.07).

**3.6.2 Sarcopenia and 30 days hospital readmission.**   The meta-analysis corresponds to the evaluation of sarcopenia as a risk factor for 30 days readmission in patients undergoing digestive tract surgery.

After reviews, 7 publications were considered that reported the presence of sarcopenia and 30 days readmission. The odds ratio (OR) values and their respective 95% confidence intervals (95% CI) were calculated.

In Fig 5, the forest plot is shown. Of the 7 publications, one found the presence of sarcopenia as a factor of 30 days readmission (p <0.05)[34]. The global OR value was 2.2 (95% CI 1.44–3.36). The measure of heterogeneity $I^2$ (Higgins heterogeneity measure) was 0%, a value considered as low heterogeneity. According to Cochran's Q heterogeneity test, the sample evidence did not allow us to reject the null hypothesis of non-heterogeneity (p = 0.790).

The publication bias was assessed using the funnel plot (Fig 6). The graph represents each of the studies with the value of association and effect measures.

## 4. Discussion

Sarcopenia is a pathological condition of increasing importance and increasingly prevalence, given the aging population and greater access to diagnostic methods. Patients at risk of developing this disease include the elderly, patients with chronic-degenerative diseases, and cancer patients. It is associated with an increased risk of adverse outcomes, including falls, fractures, postoperative complications, and mortality [40].

With technological advances and better procedural safety, more patients are eligible for gastrointestinal surgeries of various types, including resections of tumors and liver transplants, so that the study of factors that imply changes in the surgical risk of these patients is of great value for medical practice.

**Sarcopenia and Hospital Readmission (30 days)**

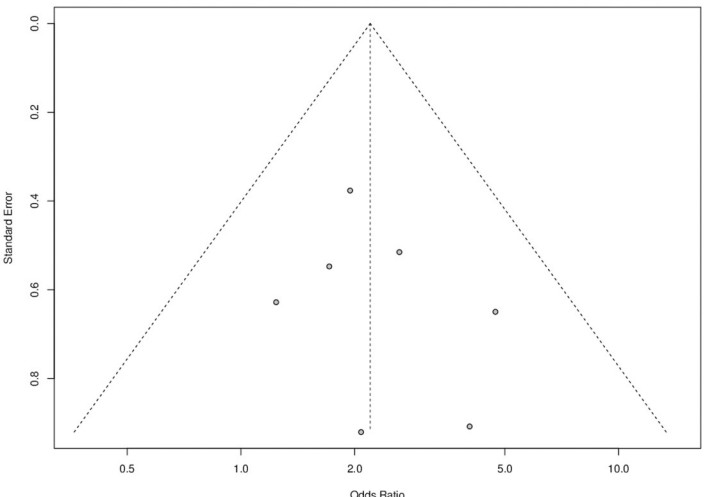

Fig 5. **Forest plot of the publications analysed regarding sarcopenia and hospital readmission (30 days).**

Fig 6. **Publication bias for studies that assessed sarcopenia as a factor associated with hospital readmission (30 days).**

The prognostic influence of sarcopenia in surgical patients has received great attention in recent years, as attested by the profusion of articles and meta-analyses carried out investigating this relation[44–46]. Hajibandeh et al. (2019) analyzed the effect of sarcopenia on postoperative mortality in patients undergoing emergency abdominal surgeries, identifying longer hospital stays, greater need for an ICU, and increased mortality within 30 days (RR 2.15) and 1 year after surgery (RR 1.97)[45]. The presence of sarcopenia was also associated with increased length of hospital stay after pancreatic surgery in the meta-analysis by Ratnayake et al [46], although a higher risk of postoperative complications has not been identified, which was partially attributed to the heterogeneity of the data. Lanza et al. (2020) investigated patients undergoing bland transarterial embolization for unresectable hepatocellular carcinoma and found

out that sarcopenia can be used as a predictor of survival, with an HR = 2.22 for reduced survival[44]. However, the majority of these studies used the term "sarcopenia" considered only reduction of muscle mass evaluated by tomography [45–47]. In specific populations, such as cirrhotic patients in a transplant waiting list, there are studies that have diagnosed sarcopenia by associating a measure of muscle strength with a measure of muscle mass—for example, handgrip strength assessed by dynamometer and image assessed by DEXA [48]. In the case of elective abdominal surgeries, virtually 100% of patients undergo abdominal CT scans before surgery; these images, combined with the simple measurement of muscle strength using a dynamometer, make it possible to carry out the preoperative diagnosis of sarcopenia. We believe that the association of imaging criteria with functional criteria increases the specificity of the diagnosis, making the prevalence data more robust and less dependent on cutoff points for muscle mass, in addition to enabling the allocation of preoperative resources to patients with greater potential of benefit. More recent studies have recognized the loss of muscle strength as a better predictive factor of adverse results than just a reduction in muscle mass [49,50], and international guidelines recommend that the diagnosis of this condition be performed confirming loss of muscle strength coupled with the reduction of lean mass, using as a parameter of severity the presence or absence of reduced muscle performance [40].

Recent systematic reviews and meta-analysis that did not use the established international criteria for sarcopenia demonstrated no influence of sarcopenia [46,51] or an odds ratio value for complications much lower than the one found in our analysis [52–54]. Comparing those results with ours clearly shows that, without proper diagnose of sarcopenia, its impact is underestimated. The use of radiological and clinical criteria, as suggested in our study, creates a more homogenous group, being a better predictor of the real impact of sarcopenia.

Therefore, in this review and meta-analysis, we seek to analyse the impact of sarcopenia in the postoperative period of gastrointestinal surgeries. To our knowledge, this is the first review that selected only articles that used functional and imaging criteria to define the diagnosis of sarcopenia, in accordance with international guidelines.

With these criteria, we found that the presence of sarcopenia is associated with an increased risk of postoperative complications (OR 3.01 [95% CI 2.55–3.55]) as well as an increase in hospital 30 days readmission (OR 2.2 [95% CI 1.44–3.36]), in patients undergoing gastrointestinal surgery. Increased complications in gastric cancer surgery and esophagectomy have also been identified. Studies have reduced heterogeneity, which increases confidence in the results. These findings confirm other studies that demonstrate sarcopenia as a factor with a worse prognosis, and may have important implications for the assessment of surgical risk and preoperative therapeutic management of these patients. For example, Yamamoto et al. (2017) [55] reported a physical exercise and nutritional optimization program in sarcopenic patients with gastric cancer implemented before surgery. Despite the reduced number of patients and the duration of the program, it was found that four patients reversed the condition of sarcopenia in the preoperative period.

Among the limitations of this analysis, we mention the presence of studies with oriental populations only, all of which were carried out in China or Japan. Thus, the results found may not be representative of the general population, and in particular, of the western population. In addition, all studies reported oncological surgeries, further limiting the analysed population.

The analysis of the impact of sarcopenia on mortality after surgery was not possible since the majority of studies in the period analysed were zero or extremely low, both in the group of sarcopenia and in the control group.

Another possible limitation of the present study is the selection of articles for systematic review and meta-analysis. Some articles did not make clear the method of assessing sarcopenia

and, for this reason, did not enter the analysis. Moreover, another issue is the use of different cutoffs among studies, partially due to the absence of exact values to define sarcopenic patients homogenously.

Finally, considering that almost all patients undergoing elective gastrointestinal surgery will have a CT for surgical preoperative planning, adding strength measurement will have a low impact in cost and time, but significant impact on the proper diagnose of sarcopenia. This classification will give more accurate information about postoperative complications risk.

Therefore, in view of the reported data, we consider that the implementation of protocols that properly diagnose sarcopenia to be of great value in the preoperative evaluation of any patient who will undergo gastrointestinal surgery. In this sense, further studies that evaluate the impact of performing preoperative interventions (e.g. resistance exercise program, optimization of protein intake [56], vitamin D supplementation, among others) are necessary, potentially minimizing the incidence of adverse events and even postoperative mortality.

## 5. Conclusion

Sarcopenia, when properly diagnosed, is associated with an increase in late postoperative complications, as well as an increase in the number of postoperative hospital readmissions for various types of gastrointestinal surgery. We believe that any preoperative evaluation should include, in a patient at risk, tests for the diagnosis of sarcopenia and appropriate procedures to reduce its impact on the patient's health. Future studies will help to determine the impact of such interventions on the rate of postoperative complications in these patients.

## Supporting information

**S1 File.**
(DOCX)

**S1 Checklist.**
(DOC)

## Acknowledgments

The authors are thankful to Justin Axel-Berg for the English corrections, Rossana V. Mendoza López for the statistical analysis, and Fundação de Amparo à Pesquisa do Estado de São Paulo (FAPESP), Processo n˚2018/10430-6 and n˚ 2018/10423-0.

## Author Contributions

**Conceptualization:** Leonardo Zumerkorn Pipek, Carlos Guilherme Baptista, Alberto Meyer.

**Data curation:** Leonardo Zumerkorn Pipek.

**Formal analysis:** Leonardo Zumerkorn Pipek.

**Investigation:** Leonardo Zumerkorn Pipek, Carlos Guilherme Baptista, Rafaela Farias Vidigal Nascimento, Alberto Meyer.

**Methodology:** Leonardo Zumerkorn Pipek, Carlos Guilherme Baptista, Alberto Meyer.

**Project administration:** Luiz Augusto Carneiro-D'Albuquerque, Alberto Meyer, Wellington Andraus.

**Supervision:** Luiz Augusto Carneiro-D'Albuquerque, Alberto Meyer, Wellington Andraus.

**Validation:** João Victor Taba, Milena Oliveira Suzuki, Fernanda Sayuri do Nascimento, Diego Ramos Martines, Fernanda Nii.

**Visualization:** Rafaela Farias Vidigal Nascimento.

**Writing – original draft:** Leonardo Zumerkorn Pipek, Carlos Guilherme Baptista, Rafaela Farias Vidigal Nascimento.

**Writing – review & editing:** João Victor Taba, Milena Oliveira Suzuki, Fernanda Sayuri do Nascimento, Diego Ramos Martines, Fernanda Nii, Leandro Ryuchi Iuamoto.

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
