## [Decision Letter · Decision Letter 0]

14 Jul 2020

PONE-D-20-18203

THE IMPACT OF SARCOPENIA ON POSTOPERATIVE OUTCOMES AFTER GASTROINTESTINAL SURGERY: A SYSTEMATIC REVIEW AND META-ANALYSIS

PLOS ONE

Dear Dr. Meyer,

Thank you for submitting your manuscript to PLOS ONE. After careful consideration, we feel that it has merit but does not fully meet PLOS ONE’s publication criteria as it currently stands. Therefore, we invite you to submit a revised version of the manuscript that addresses the points raised during the review process.

Find below some key points raised by the reviewers. Particularly, consider referring in your discussion to the following recent articles on Sarcopenia: 

- https://doi.org/10.1371/journal.pone.0232371 

- https://doi.org/10.1097/SLA.0000000000002679

We look forward to receiving your revised manuscript.

Kind regards,

Ezio Lanza, M.D.

Academic Editor

PLOS ONE

2. In the Methods section, please indicate whether a fixed- or random effects regression model was used for the data analysis conducted in your study.

Reviewer's comments: 

Reviewer #1: This is an interesting systematic review and meta-analysis on the impact of sarcopenia on the postoperative outcomes following gastrointestinal surgeries. The clinical question is well posed and the topic is timely an appropriate. Statistics are well performed. PRISMA guidelines are strictly respected and protocol registration on an international database makes the analysis of high quality. Conclusions are supported by the results. I have some comments that could improve the manuscript.

There is an important manuscript published in the literature a couple of years ago concerning Sarcopenia as a predictor of morbidity in gastrointestinal surgical oncology. This is a systematic review and meta-analysis (Simonsen et al. Ann. Surg. 2018). I don't know why this article is not even mentioned in your manuscript. Was this missed? You definetely have to comment this manuscript in your paper as it is a major one. Yours have some significant improvements i believe so please show the reader that it is worth it to have a new one on the topic.

You have correctly mentioned that because of the absolute prevalence of Eastern studies in your analysis, your results could not be reproducible among Westerners. I think that it is worth mentioning that another issue is the use of different cutoffs among studies as it is clear from your Tables. Indeed, some authors use recognized cutoffs to divide sarcopenic and non sarcopenic patients while others use ROC curves on their populations. Both methods have limitations of course and therefore this needs to be mentioned as a limitation in your study. We currently don't have an exact cutoff to define sarcopenic patients homogenously.

Finally, of course the real novelty here is that you consider the 1 and only correct definition of sarcopenia which is made up of both muscle mass and muscle strength . The authors should emphasize this in their study. In my opinion also in the title. If you leave your title as it is, you might get the reader confused. Indeed, when i read your title to review i thought that i was about t read something with no novelty (Very similar to the one from Simonsen et al.). But indeed you have something new and important to tell. Please consider giving your title some more appeal.

Reviewer #2: This is a nice systematic review and meta-analysis to address the role of sarcopenia in the postoperative complications of gastrointestinal surgery. Authors have found that sarcopenia was significantly associated with an increase in late postoperative complications, as well as an increase in the number of postoperative hospital readmissions after gastrointestinal surgery. Thus, it is important to assess sarcopenia and provide appropriate interventions before surgery to reduce its impact on the patient's health. I think that this is a well-conducted systematic review according to the PRISM statement. However, liver transplantation should not be included in gastrointestinal surgery and the data should be removed from the analysis.

Minor point.

References 4 and 23 are overlapped.

---

## [Author Response · Author response to Decision Letter 0]

29 Jul 2020

We really appreciate all the comments, we believe our text is now substantially better.

#1: This is an interesting systematic review and meta-analysis on the impact of sarcopenia on the postoperative outcomes following gastrointestinal surgeries. The clinical question is well posed and the topic is timely an appropriate. Statistics are well performed. PRISMA guidelines are strictly respected and protocol registration on an international database makes the analysis of high quality. Conclusions are supported by the results. I have some comments that could improve the manuscript.

There is an important manuscript published in the literature a couple of years ago concerning Sarcopenia as a predictor of morbidity in gastrointestinal surgical oncology. This is a systematic review and meta-analysis (Simonsen et al. Ann. Surg. 2018). I don't know why this article is not even mentioned in your manuscript. Was this missed? You definetely have to comment this manuscript in your paper as it is a major one. 

Thank you for pointing out this article. Indeed, this was very valuable, and we added to our discussion. (Reference 52). 

Yours have some significant improvements i believe so please show the reader that it is worth it to have a new one on the topic.

We have now emphasized this in our title, abstract and in the body of the text.

You have correctly mentioned that because of the absolute prevalence of Eastern studies in your analysis, your results could not be reproducible among Westerners. I think that it is worth mentioning that another issue is the use of different cutoffs among studies as it is clear from your Tables. Indeed, some authors use recognized cutoffs to divide sarcopenic and non sarcopenic patients while others use ROC curves on their populations. Both methods have limitations of course and therefore this needs to be mentioned as a limitation in your study. We currently don't have an exact cutoff to define sarcopenic patients homogenously.

We totally agree with you and added this limitation to our discussion.

Finally, of course the real novelty here is that you consider the 1 and only correct definition of sarcopenia which is made up of both muscle mass and muscle strength . The authors should emphasize this in their study. In my opinion also in the title. If you leave your title as it is, you might get the reader confused. Indeed, when i read your title to review i thought that i was about t read something with no novelty (Very similar to the one from Simonsen et al.). But indeed you have something new and important to tell. Please consider giving your title some more appeal.

Thank you very much for this insight, we changed our title to show the novelty of our study. 

Reviewer #2: This is a nice systematic review and meta-analysis to address the role of sarcopenia in the postoperative complications of gastrointestinal surgery. Authors have found that sarcopenia was significantly associated with an increase in late postoperative complications, as well as an increase in the number of postoperative hospital readmissions after gastrointestinal surgery. Thus, it is important to assess sarcopenia and provide appropriate interventions before surgery to reduce its impact on the patient's health. I think that this is a well-conducted systematic review according to the PRISM statement. However, liver transplantation should not be included in gastrointestinal surgery and the data should be removed from the analysis.

We removed this study (Harimoto et al. 2017) and changed all figures and tables. The final results of our meta analyses barely changed. We didn’t have to change any conclusion about our analyses. 

Minor point.

References 4 and 23 are overlapped.

We corrected this and also included new references that were suggested (ref 44 and 52).

---

## [Editor Report · Decision Letter 1]

3 Aug 2020

THE IMPACT OF PROPERLY DIAGNOSED  SARCOPENIA ON POSTOPERATIVE OUTCOMES AFTER GASTROINTESTINAL SURGERY: A SYSTEMATIC REVIEW AND META-ANALYSIS

PONE-D-20-18203R1

Dear Dr. Meyer,

We’re pleased to inform you that your manuscript has been judged scientifically suitable for publication and will be formally accepted for publication once it meets all outstanding technical requirements.

Kind regards,

Ezio Lanza, M.D.

Academic Editor

PLOS ONE

---

## [Editor Report · Acceptance letter]

5 Aug 2020

PONE-D-20-18203R1 

THE IMPACT OF PROPERLY DIAGNOSED  SARCOPENIA ON POSTOPERATIVE OUTCOMES AFTER GASTROINTESTINAL SURGERY: A SYSTEMATIC REVIEW AND META-ANALYSIS 

Dear Dr. Meyer:

I'm pleased to inform you that your manuscript has been deemed suitable for publication in PLOS ONE. Congratulations! Your manuscript is now with our production department. 

Kind regards, 

on behalf of

Dr. Ezio Lanza 

Academic Editor

PLOS ONE